# Poor treatment outcome and associated risk factors among patients with isoniazid mono-resistant tuberculosis: A systematic review and meta-analysis

**Ayinalem Alemu**[1,2]*, **Zebenay Workneh Bitew**[3], **Getu Diriba**[1], **Getachew Seid**[1,2], **Shewki Moga**[1], **Saro Abdella**[1], **Emebet Gashu**[4], **Kirubel Eshetu**[5], **Getachew Tollera**[1], **Mesay Hailu Dangisso**[1], **Balako Gumi**[2]

**1** Ethiopian Public Health Institute, Addis Ababa, Ethiopia, **2** Aklilu Lemma Institute of Pathobiology, Addis Ababa University, Addis Ababa, Ethiopia, **3** St. Paul's Hospital Millennium Medical College, Addis Ababa, Ethiopia, **4** Addis Ababa Health Bureau, Addis Ababa, Ethiopia, **5** USAID Eliminate TB Project, Management Sciences for Health, Addis Ababa, Ethiopia

* ayinalemal@gmail.com, ayinalem.alemu@aau.edu.et

**Data Availability Statement:** All relevant data are within the paper and its Supporting Information files.

## Abstract

### Background

To date, isoniazid mono-resistant tuberculosis (TB) is becoming an emerging global public health problem. It is associated with poor treatment outcome. Different studies have assessed the treatment outcome of isoniazid mono-resistant TB cases, however, the findings are inconsistent and there is limited global comprehensive report. Thus, this study aimed to assess the poor treatment outcome and its associated risk factors among patients with isoniazid mono-resistant TB.

### Methods

Studies that reported the treatment outcomes and associated factors among isoniazid mono-resistant TB were searched from electronic databases and other sources. We used Joana Briggs Institute critical appraisal tool to assess the study's quality. We assessed publication bias through visual inspection of the funnel plot and confirmed by Egger's regression test. We used STATA version 17 for statistical analysis.

### Results

Among 347 studies identified from the whole search, data were extracted from 25 studies reported from 47 countries. The pooled successful and poor treatment outcomes were 78% (95%CI; 74%-83%) and 22% (95%CI; 17%-26%), respectively. Specifically, complete, cure, treatment failure, mortality, loss to follow-up and relapse rates were 34%(95%CI; 17%-52%), 62% (95%CI; 50%-73%), 5% (95%CI; 3%-7%), 6% (95%CI; 4%-8%), 12% (95%CI; 8%-17%), and 1.7% (95%CI; 0.4%-3.1%), respectively. Higher prevalence of pooled poor treatment outcome was found in the South East Asian Region (estimate; 40%, 95%C; 34%-45%), and African Region (estimate; 33%, 95%CI; 24%-42%). Previous TB treatment (OR;

**Funding:** The author(s) received no specific funding for this work.

**Competing interests:** The authors have declared that no competing interests exist.

1.74, 95%CI; 1.15–2.33), having cancer (OR; 3.53, 95%CI; 1.43–5.62), and being initially smear positive (OR; 1.26, 95%CI; 1.08–1.43) were associated with poor treatment outcome. While those patients who took rifampicin in the continuation phase (OR; 0.22, 95%CI; 0.04–0.41), had extrapulmonary TB (OR; 0.70, 95%CI; 0.55–0.85), and took second-line injectable drugs (OR; 0.54, 95%CI; 0.33–0.75) had reduced risk of poor treatment outcome.

## Conclusion

Isoniazid mono-resistant TB patients had high poor treatment outcome. Thus, determination of isoniazid resistance pattern for all bacteriologically confirmed TB cases is critical for successful treatment outcome.

**PROSPERO registration number:** CRD42022372367

## Introduction

Tuberculosis (TB) is causing a huge public health impact being the second cause of mortality among infectious diseases. There were 9.9 million TB cases and more than 1.5 million deaths due to TB in 2020 [1]. The efforts for the prevention and control of TB becomes challenging due to the emergence of drug resistant TB mainly with respect to treatment outcome. Drug-resistant TB is associated with poor treatment outcome [1, 2]. Based on the 2021 global TB report, the global successful treatment outcome among drug susceptible and Multi-drug resistant TB (MDR-TB)/ Rifampicin resistant TB (RR-TB) cases were 86% and 59%, respectively [1]. Drug resistant TB have different categories including mono-resistant TB. When TB is caused by *Mycobacterium tuberculosis* strains which are resistant only to one anti-TB drug it is called mono-resistant TB and isoniazid mono-resistant TB is among the categories [1, 2].

The world health organization (WHO) through the END TB Strategic document recommends calls for the early TB diagnosis drug sensitivity testing (DST) [3]. The drug resistance pattern should be determined for all bacteriologically confirmed TB cases to put patients on the right treatment for successful treatment outcome and to prevent the emergence of additional drug-resistance. Even though, there are improvements in the recent years, this becomes difficult in many TB endemic low and middle-income countries having resource limitations. To date, due to the implementation of Xpert MTB/RIF assay many countries reported RR-TB to the WHO [1, 2]. In this assay, the resistance profile for the other potent anti-TB drug isoniazid is unknown that might have made the isoniazid mono-resistant TB cases to be less reported and be treated as drug susceptible TB [2]. However, about 11% of TB patients worldwide are estimated to have isoniazid resistant, rifampicin susceptible TB [2].

Studies conducted in different settings indicated that isoniazid mono-resistant TB is a problem in different countries [4–8]. The incidence of isoniazid mono-resistant TB is increasing and it is higher than RR-TB globally [9]. In addition, studies revealed that those isoniazid mono-resistant TB cases had higher rate of poor treatment outcome compared to the drug-susceptible TB cases [10–13]. There are studies that assessed the treatment outcome of isoniazid mono-resistant TB cases [4–7, 10–30], however, the findings are inconsistent. In addition, there is no comprehensive report at the global level. Thus, this study aimed to assess the poor treatment outcome and the associated risk factors among patients with isoniazid mono-resistant TB.

## Methods

### Protocol registration

The protocol for this study is registered on the international prospective register of systematic reviews (PROSPERO) with a registration number CRD42022372367.

### Information source and search strategy

This study was developed following the Preferred Reporting Items for Systematic Reviews and Meta-Analyses (PRISMA) reporting checklist [31] (**S1 Table**). Article searching was conducted systematically from the electronic databases including PubMed, CINAHL, Global Health, Global Health Medicus and Environment Index. In addition, our search extends to other grey literature sources such as Google and Google Scholar. The search was conducted up to 20 November 2022 for studies published in English language. Two authors (AA, EG) have conducted the article searching independently. The third author (ZWB) managed the inconsistencies arose between the two authors. The search was conducted using the keywords; isoniazid mono-resistant tuberculosis, treatment outcome and risk factors/determinants. The Boolean operators OR and AND were used accordingly. The search string for PubMed was ("Treatment Outcome"[MeSH Terms] OR (("poverty"[MeSH Terms] OR "poverty"[All Fields] OR "poor"[All Fields]) AND ("Treatment Outcome"[MeSH Terms] OR ("treatment"[All Fields] AND "outcome"[All Fields]) OR "Treatment Outcome"[All Fields])) OR ("Treatment Outcome"[MeSH Terms] OR ("treatment"[All Fields] AND "outcome"[All Fields]) OR "Treatment Outcome"[All Fields])) AND (("isoniazid"[MeSH Terms] OR "isoniazid"[All Fields] OR "isoniazide"[All Fields]) AND "mono-resistant"[All Fields]) (**S2 Table**).

### Study selection procedure

We have followed a step-wise approach to select the eligible studies. Primarily, all the studies identified from the whole search were exported to EndNote X8 citation manager, and we have removed the duplicates. In the next step, we have screened the articles by title and abstract. Then, full-text assessment was conducted for the remaining articles. Finally, we have included the articles that passed the full-text review in the final analysis. The article selection procedure was conducted by two independent authors (GD, GS) using pre-defined criteria that considered study subjects, study designs, quality, and outcome (**Fig 1**).

### PICOS criteria

Participants: Isoniazid mono-resistant tuberculosis patients
 Intervention: Anti-TB treatment
 Comparator: Successful treatment outcome
 Outcome: Poor treatment outcome
 Study design: Observational studies.
 Study setting: Any setting in any country across the globe

### Inclusion and exclusion criteria

Studies that reported either TB treatment outcome or risk factors of poor treatment outcome or both in patients with isoniazid mono-resistant TB were included in the study. There was no restriction on entering the study in terms of sample size. The exclusion criteria were review studies, and not differentiated the target population.

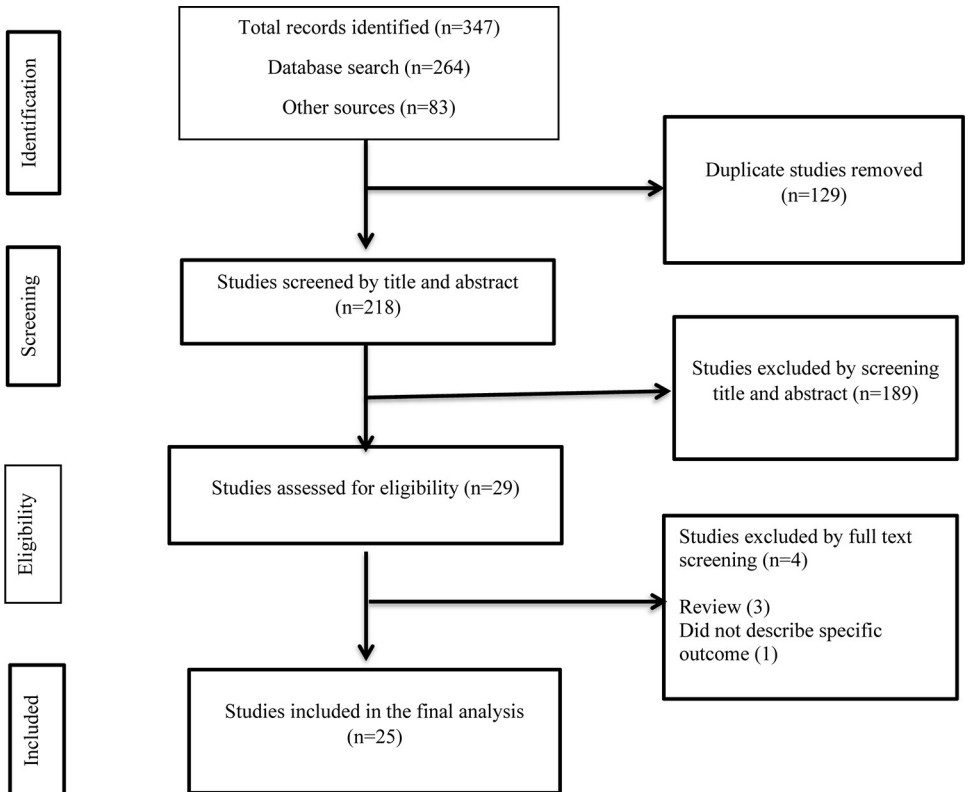

**Fig 1. Flowchart describing the selection of studies for the systematic review and meta-analysis of poor treatment outcome and its associated factors among patients with isoniazid mono-resistant tuberculosis.**

## Data extraction

Data were extracted from the articles included in the final analysis using Microsoft Excel 2016 spreadsheet. The extracted data included; primary author name, publication year, country, data collection period, study design, data collection time (prospective vs retrospective), study setting/place, age of study participants, number of study participants, number having successful (completed, cured) and poor treatment outcomes (mortality, treatment failure, loss to follow-up), number of relapse in successfully treated cases, and factors associated with poor treatment outcomes. Data were extracted by two independent authors (AA, ZWB), and the third author (GD) managed the inconsistencies that arose between the two authors.

## Risk of bias (quality) assessment of included studies

We have evaluated the methodological reputability and quality of the findings of the included studies using the Joanna Briggs Institute (JBI) critical appraisal tools for observational studies [32]. Two independent authors (GS, KE) conducted the quality assessment, and the third author (ZWB) resolved the inconsistencies. The checklist for cross-sectional, case control and cohort studies consists of 8, 10, and 11 indicators, respectively. Each indicator was equally scored and summed up to give 100%. The quality of the studies was scored to have high, medium and low quality if the overall quality score was >80%, 60–80% and <60%, respectively (S3 Table). The presence of publication bias was explored through visual evaluation of the funnel plot such that asymmetry of the funnel plot indicated the presence of publication bias.

Furthermore, we have conducted egger's regression test to confirm the presence of publication bias (P<0.05).

## Outcomes

The primary outcome of this study was the treatment outcomes such as the successful and poor treatment outcomes along with different categories among patients with isoniazid mono-resistant tuberculosis. The secondary outcomes were the factors that associated with poor treatment outcomes in those patients.

## Operational definition

The operational definition for isoniazid mono-resistant tuberculosis was based on the WHO definition. This type of tuberculosis is caused by *Mycobacterium tuberculosis* strains that are resistant to isoniazid but susceptible to rifampicin confirmed in vitro [33]. The definitions for the treatment outcomes is based on the WHO classification of TB treatment outcomes as described in the guideline [34].

## Ethical approval and consent to participate

Since this study is based on a review of published articles, ethical approval is not mandatory. The protocol is registered on PROSPERO.

## Data synthesis and statistical analysis

The pooled estimates of successful and poor treatment outcomes among patients with isoniazid mono-resistant TB was determined with its 95%CI by assuming the true effect size varies between studies. The pooled estimate for successful and poor treatment outcomes were determined as the ratio of numbers of isoniazid mono-resistant TB patients with successful and poor treatment outcomes to the total treated isoniazid mono-resistant TB patients, respectively. Besides, the pooled OR along with 95%CI was estimated for each factor to determine the factors associated with poor treatment outcomes. We have also performed a stratified analysis. We presented the data using the forest plot. The heterogeneity among the studies was assessed using the $I^2$ heterogeneity test and a value above 50% indicated the presence of substantial heterogeneity among studies [35, 36]. We have performed bi-variable and multi-variable meta-regression to assess the association of study year and sample size on poor treatment outcome. To assess the presence of publication bias, the funnel plot was inspected visually and Egger's regression test was conducted. For those parameters that had a publication bias (P<0.05) in the Egger's regression test [37, 38], we have performed a trim and fill analysis to adjust the publication bias. The statistical analysis was conducted using STATA version 17.

## Results

### Characteristics of included studies

From the whole search, we identified 347 studies and after removing 129 duplicates, 218 were screened by title and abstract. At this stage, 189 studies were excluded and the remaining 29 studies were screened by full text review. Finally, 25 studies were included in this study [4–7, 10–30]. These studies were reported from five continents and from all the six WHO regions. Accordingly, the most frequent number of studies were reported from Asia with 11 studies followed by North America (5 studies), Africa (4 studies), Europe (3 studies), and South America (2 studies). Per WHO regional classification, relatively higher number of studies were reported from the Region of Americas (AMR) with 7 studies. The frequencies of studies in the other

regions were; West Pacific Region (WPR) (5 studies), African Region (AFR) (4 studies), European Region (EUR) (4 studies), South Eastern Asian Region (SEAR) (3 studies), and Eastern Mediterranean Region (EMR) (2 studies). The studies were reported from 47 countries and a maximum of two studies were reported from a single country (South Africa, Taiwan, China, Portugal, USA, Canada, India, and Peru). A single study conducted in Europe comprises data collected from 31 countries [20] that made the number of countries included in the current systematic review and meta-analysis study to be 47 in number.

The studies were published from 2009 [15, 29] to 2022 [28]. The data collection period for most of the studies were after 2000 except two studies where the data collection period was from October 1992 to October 2005 for one study [15] and from 1995 to 2010 for the other study [11]. In the majority of the studies (88%, 22), data were collected retrospectively. The data in these studies were collected either from a health facility or from the national surveillance data registry database (**Table 1**).

## Pooled treatment outcomes among isoniazid mono-resistant tuberculosis patients

In the current study, we extracted data to estimate the pooled prevalence of successful treatment outcome including cure rate and treatment completion rate, poor treatment outcome including death rate, treatment failure rate and loss to follow-up, relapse after successful treatment outcome, and factors associated with poor treatment outcome among patients with isoniazid mono-resistant tuberculosis.

Data were extracted from 24 and 23 studies to estimate the pooled prevalence of successful treatment outcome and poor treatment outcome, respectively. The largest sample size was 6796 in a study that comprises 31 European countries [20], while the smallest sample size was 9 in a study conducted in Saudi Arabia [21]. Among the studies, 11 studies had a sample size below 100 while the remaining studies had a sample size of 132 and above.

Based on data collected from 24 studies comprising 10, 698 isoniazid mono-resistant TB patients, 8606 had successful treatment outcome that gave a pooled estimate of 78% (95%CI; 74–83, $I^2$; 94.02%) (**Fig 2**). The symmetry of the funnel plot (**Fig 3**) and the statistical insignificance of the egger's regression test showed there is no publication bias (P = 0.080). Specifically, the pooled treatment completed and cured rate among isoniazid mono-resistant TB patients were 34% (95%CI; 17–52, $I^2$; 99.26%) (**S1** and **S2** Figs) and 62% (95%CI; 50–73, $I^2$; 96.91%) (**S3** and **S4** Figs), respectively. Based on the WHO regional classification, the pooled prevalence of successful treatment outcome from the highest to lowest pooled estimate were; AMR (estimate; 84%; 95%CI; 78–90, $I^2$; 87.66%), EUR (estimate; 84%; 95%CI; 77–91, $I^2$; 91.21%), WPR (estimate; 82%; 95%CI; 77–86, $I^2$; 64.67%), EMR (estimate; 75%; 95%CI; 44–106, $I^2$; 73.41%), AFR (estimate; 67%; 95%CI; 58–76, $I^2$; 74.28%), and SEAR (estimate; 62%; 95%CI; 56–69, $I^2$; 13.74%) (**Fig 2**) (**Table 2**).

The poor treatment outcome was estimated from 23 studies having 10,670 isoniazid mono-resistant TB patients. From these individuals, 2084 had poor treatment outcome that yield a pooled estimate of 22% (95%CI; 17–26, $I^2$; 94.08%) (**Fig 4**). The egger's regression test showed there is no publication bias (P = 0.107) (**Fig 5**). Specifically, the pooled treatment failure, mortality and loss to follow-up rates were 5% (95%CI; 3–7, $I^2$; 93.97%) (**S5** and **S6** Figs), 6% (95% CI; 4–8, $I^2$; 88.73%) (**S7** and **S9** Figs), and 12% (95%CI; 8–17, $I^2$; 96.58%) (**S9** and **S10** Figs), respectively. Based on the WHO regional classification, the pooled prevalence of poor treatment outcome from the highest to lowest pooled estimate was; SEAR (estimate; 40%; 95%C; I34-45, $I^2$; 0.00%), AFR (estimate; 33%; 95%CI; 24–42, $I^2$; 74.28%), EMR (estimate; 25%; 95% CI; -0.06–56, $I^2$; 73.41%), WPR (estimate; 18%; 95%CI; 14–23, $I^2$; 64.63%), EUR (estimate;

**Table 1. Characteristics of individual studies on the poor treatment outcome and associated risk factors among patients with isoniazid mono-resistant tuberculosis included in the current systematic review and meta-analysis.**

| Author year | Publication year | Country | Study period | Study design | Data collection time | Study setting | Age group | Sample size | Successful outcome | | Poor outcome | |
|---|---|---|---|---|---|---|---|---|---|---|---|---|
| | | | | | | | | | N | % | N | % |
| Chien et al., 2014 | 2014 | Taiwan | January 2004 to October 2011 | Retrospective cohort study | Retrospectively | Four hospitals in northern, central, southern and eastern Taiwan | All age groups (Median age was 64 years) | 395 | 328 | 83.04 | 67 | 16.96 |
| Bachir et al., 2021 | 2021 | France | January 1, 2016 to December 31, 2017 | Multicenter case-control study | Retrospectively | University hospitals of Paris, Lille, Caen and Strasbourg | Median age was 35 years | 97 | 75 | 77.32 | 22 | 22.68 |
| Cattamanchi et al., 2009* | 2009 | USA | October 1992 to October 2005 | Retrospective cohort study | Retrospectively | San Francisco Department of Public Health Tuberculosis Control Section | Median age was 47 years | 137 | - | - | - | - |
| Kwak et al., 2020 | 2020 | South Korea | January 2005 to December 2018 | Retrospective record review | Retrospectively | South Korean tertiary referral hospital | ≥18 years | 195 | 164 | 84.10 | 31 | 15.90 |
| Binkhamis et al., 2021 | 2021 | Saudi Arabia | May 2015 and April 2019 | Cross-sectional analytical study | Retrospectively | King Khalid University Hospital | All age groups (range:1–90 years) | 9 | 5 | 55.56 | 4 | 44.44 |
| Murwira, et al., 2020 | 2020 | Zimbabwe | March 2017 and December 2018 | Retrospective cohort study | Retrospectively | National TB Reference Laboratory (NTBRL) in Bulawayo City and National TB programme | All age groups (Median age was 36 years, Interquartile range, was 29–45 years) | 31 | 25 | 80.65 | 6 | 19.35 |
| Chierakul et al., 2014 | 2014 | Thailand | July 2009 and July 2011 | Retrospective cohort study | Retrospectively | Siriraj Hospital | > 15 years | 28 | 20 | 71.43 | - | - |
| Jacobson et al., 2011 | 2011 | South Africa | 28 November 2000 to 28 May 2009 | Retrospective cohort study | Retrospectively | 22 clinics in the rural Cape Winelands East and Overberg Districts, Western Cape Province | All age groups (range:11–67 years) | 151 | 101 | 66.89 | 50 | 33.11 |
| Garcia et al., 2018 | 2018 | Peru | January 2012 and December 2014 | Cross-sectional study | Retrospectively | National registry of drug-resistant tuberculosis | All age groups | 947 | 731 | 77.19 | 216 | 22.81 |
| Karo et al., 2018 | 2018 | 31 European countries | 2002 to 2014 | Observational study | Retrospectively | European Surveillance System (TESSy) | All age groups (Median age was 41 years) | 6796 | 5611 | 82.56 | 1185 | 17.44 |
| Saldaña et al., 2016 | 2016 | Mexico | 1995 to 2010 | Prospective cohort study | Prospectively | 12 municipalities in the Orizaba Health Jurisdiction in Veracruz State | > 15 years | 85 | 64 | 75.29 | 21 | 24.71 |
| Villegas et al., 2016 | 2016 | Peru | March 2010 to December 2011 | Prospective cohort study | Prospectively | 34 health facilities in a northern district of Lima | All age groups | 82 | 63 | 76.83 | 19 | 23.17 |

*(Continued)*

**Table 1.** (Continued)

| Author year | Publication year | Country | Study period | Study design | Data collection time | Study setting | Age group | Sample size | Successful outcome | | Poor outcome | |
|---|---|---|---|---|---|---|---|---|---|---|---|---|
| | | | | | | | | | N | % | N | % |
| Edwards et al., 2020 | 2020 | Canada | 2007 to 2017 | Retrospective cohort study | Retrospectively | One of three centralized comprehensive clinics in the province of Alberta | Median age was 37 years | 98 | 90 | 91.84 | 8 | 8.16 |
| Wang et al., 2014 | 2014 | Taiwan | 2006 January to 2007 December | Retrospective cohort study | Retrospectively | Chang Gung Memorial Hospital | All age groups | 134 | 114 | 85.07 | 20 | 14.93 |
| Sayfutdinov et al., 2021 | 2021 | Uzbekistan | 2017 to 2018 | Retrospective cohort study | Retrospectively | Two regions of Uzbekistan (Fergana and Bukhara) | All age groups | 132 | 105 | 79.55 | 27 | 20.45 |
| der Heijden et al., 2017 | 2017 | South Africa | 2000 to 2012 | Longitudinal study | Retrospectively | Prince Cyril Zulu Communicable Diseases Centre (PCZCDC) | All age groups (Median age was 34 years) | 405 | 235 | 58.02 | 170 | 41.98 |
| Romanowski et al., 2017 | 2017 | Canada | 2002 to 2014 | Retrospective record review | Retrospectively | BC Centre for Disease Control (BCCDC) | All age groups (Median age was 46 years) | 152 | 140 | 92.11 | 12 | 7.89 |
| Santos et al., 2018 | 2018 | Portugal | 01 January 2008 to 31 December 2014 | Retrospective record review | Retrospectively | National-Tuberculosis-Surveillance-System (SVIG-TB) | All age groups (Median age was 44 years) | 232 | 210 | 90.52 | 22 | 9.48 |
| Shao et al., 2020 | 2020 | China | 2013 to 2018 | Retrospective cohort study | Retrospectively | Four national DR-TB surveillance sites of Jiangsu Province | All age groups (Median age was 48 years) | 63 | 52 | 82.54 | 11 | 17.46 |
| Kuaban et al., 2020 | 2020 | Cameroon | January 2012 to March 2015 | Retrospective record review | Retrospectively | In all the TB diagnostic and treatment centres (DTCs) in four regions of Cameroon namely the North West, South West, West, and Littoral regions | All age groups (range: 17–79 years) | 45 | 32 | 71.11 | 13 | 28.89 |
| Salindri et al., 2018 | 2018 | USA | 2009 to 2014 | Retrospective cohort study | Retrospectively | Georgia State Electronic Notifiable Disease Surveillance System (SENDSS) | ≥15 years | 140 | 124 | 88.57 | 16 | 11.43 |
| Nagar et al., 2022 | 2022 | India | January 2019 to December 2020 | Retrospective record review | Retrospectively | Ahmedabad city from Ni-kshay, an online web-based portal | ≥18 years | 243 | 144 | 59.26 | 99 | 40.74 |
| Tabarsi et al., 2009 | 2009 | Iran | 2003 to 2005 | Prospective cohort study | Prospectively | Masih Daneshvari Hospital | All age groups | 42 | 37 | 88.10 | 5 | 11.90 |
| Chunrong et al., 2020 | 2020 | China | January 2016 to January 2019 | Retrospective record review | Retrospectively | Shenzhen's drug-resistant TB project | All age groups (17–75 years) | 144 | 102 | 70.83 | 42 | 29.17 |

*(Continued)*

**Table 1.** (Continued)

| Author year | Publication year | Country | Study period | Study design | Data collection time | Study setting | Age group | Sample size | Successful outcome | | Poor outcome | |
|---|---|---|---|---|---|---|---|---|---|---|---|---|
| | | | | | | | | | N | % | N | % |
| Garg et al., 2019 | 2019 | India | January 1 to December 31, 2017 | Retrospective record review | Retrospectively | At the nodal DRTB centre, Department of Pulmonary Medicine, Government Medical College and Hospital, Chandigarh | All age groups | 52 | 34 | 65.38 | 18 | 34.62 |

"-"; Not specifically indicated

* the study only indicated the treatment completion rate the total successful treatment outcome including the cured cases and the poor treatment outcome (failure, death and lost to follow-up) are not indicated in the study.

17%; 95%CI; 11–22, $I^2$; 85.06%), and AMR (estimate; 16%; 95%CI; 10–22, $I^2$; 87.75%) (**Fig 4**) (**Table 2**).

## Pooled prevalence of relapse among successfully treated isoniazid mono-resistant tuberculosis patients

In this study, we have also assessed the relapse rate among isoniazid mono-resistant TB patients who had successful treatment outcome. We extracted data from eight studies comprising 970 successfully treated isoniazid mono-resistant TB cases. From these individuals, 28 developed relapse. The relapse period started from treatment completion and extends up to two years after treatment. Based on the random-effects model, the pooled prevalence of relapse among successfully treated isoniazid mono-resistant TB cases was 1.7% (95%CI; 0.4–3.1, $I^2$; 44.58%) (**Fig 6**).

## Meta-regression

Besides, we have conducted a meta-regression analysis to assess the effect of sample size and publication year on the heterogeneity among studies that reported poor treatment outcome among isoniazid mono-resistant TB patients. The multivariable meta-regression model revealed that sample size (P = 0.713) and publication year (P = 0.464) did not significantly affected heterogeneity among studies (**Table 3**).

## Risk factors of poor treatment outcome in isoniazid mono-resistant tuberculosis patients

In the current study, we assessed the risk factors associated with poor treatment outcome in isoniazid mono-resistant TB patients. We have performed the pooled estimate for the factors reported at least by two studies. We have estimated the pooled OR for 19 variables. The risk factors analyzed included demographic (sex, age group), smoking status, clinical factors such as having co-morbidities including diabetes, cancer, end-stage renal failure, and HIV, presence of cavity lesion in the chest radiograph, type of TB (extra-pulmonary vs pulmonary), initial smear status (smear positive vs smear negative), culture conversion after 2 months, drug-resistance level of isoniazid (high level vs low-level), and per taking different anti-TB drugs during

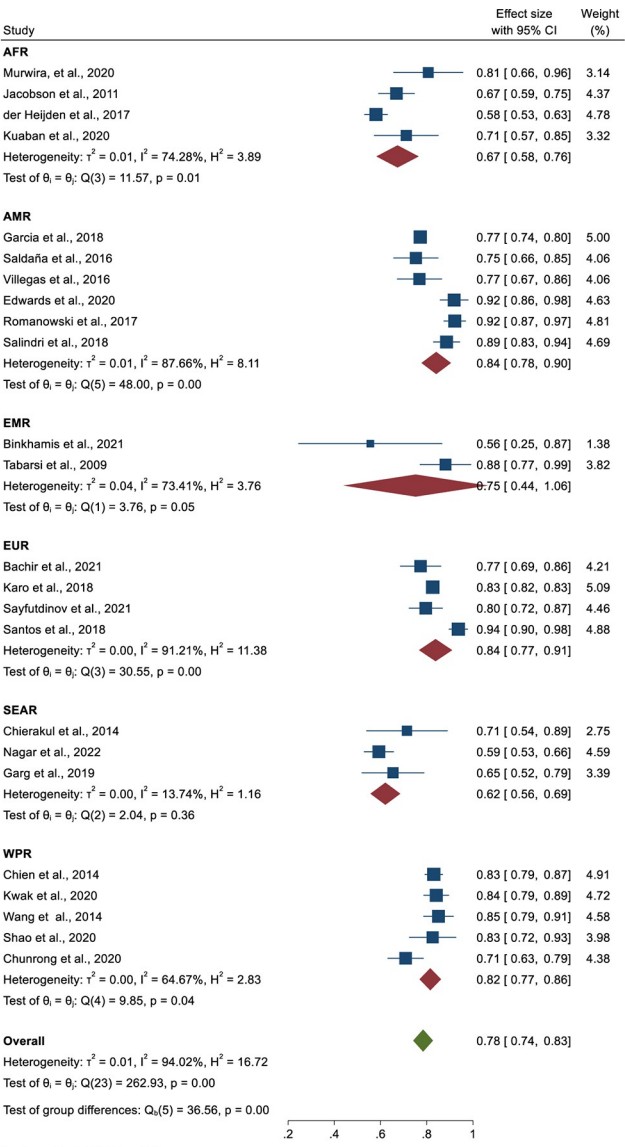

τ2; Tau (between-study variance), I² ; I-squared heterogeneity statistic (variability between studies), H² ; H-squared

heterogeneity statistic (variability between studies).

**Fig 2. Forest plot for the pooled successful treatment outcome rate among patients with isoniazid mono-resistant tuberculosis.**

initiation phase isoniazid (INH), streptomycin (STR), fluoroquinolones (FLQ), second-line injectable drugs (SLIDs) and continuation phase (rifampicin (RIF), (pyrazinamide (PZA)).

Statistically significant association was found for previous TB history (pooled OR; 1.74; 95%CI; 1.15–2.33, I² ; 45.10%) (**Fig 7**), having cancer, (pooled OR; 3.53; 95%CI; 1.43–5.62, I² ; 0.00%) (**S11 Fig**), initially smear positive (pooled OR; 1.26, 95%CI; 1.08–1.43, I² ; 2.13%) (**Fig 8**), taking RIF in the continuation phase (pooled OR; 0.22, 95%CI; 0.04–0.41, I² ; 0.00%) (**S12 Fig**), having EPTB (pooled OR; 0.70, 95%CI; 0.55–0.85, I² ; 0.00%) (**S13 Fig**), and taking SLIDs (pooled OR; 0.54, 95%CI; 0.33–0.75, I² ; 0.00%) (**S14 Fig**). Accordingly, individuals with previous TB treatment history had 1.74 times the odds to had poor treatment outcome compared to

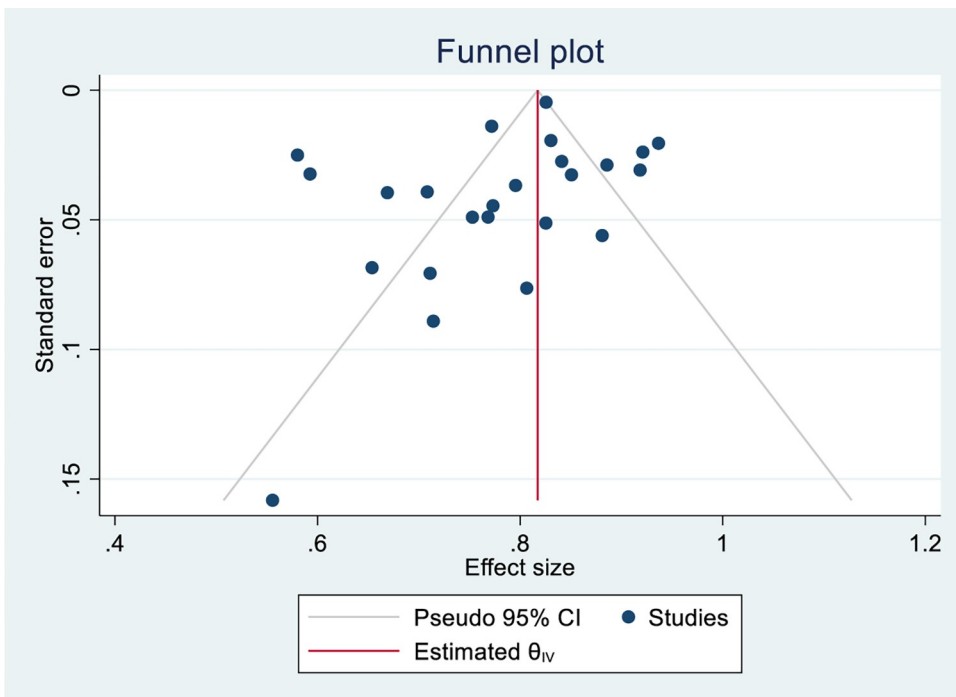

**Fig 3. Funnel plot for the pooled successful treatment outcome rate among patients with isoniazid mono-resistant tuberculosis.**

new patients. Those patients who had cancer had 3.53 times the odds to develop poor treatment outcome compared to the counterparts. In addition, those patients who were smear positive initially had 1.26 times the odds to develop poor treatment outcome compared to those having smear negative TB initially. Patients who took RIF in the continuation phase had 78% reduced risk to have poor treatment outcome compared to their counterparts. Furthermore, those who took SLIDs had 45% reduced risk to have poor treatment outcome compared to their counterparts. Besides, those patients with EPTB had 30% reduced risk of poor treatment outcomes compared to those who had pulmonary TB (**Table 2**).

Statistically significant association was not found for being male (pooled OR; 1.34, 95%CI; 0.90–1.77, $I^2$; 43.67%) (**S15 Fig**), older age (pooled OR; 0.97, 95%CI; 0.62–1,32, $I^2$;85.56%) (**S16 Fig**), being smoker (pooled OR; 95%CI; 0.89–4.20, $I^2$; 13.69%) (**S17 Fig**), having DM (pooled OR; 1.16, 95%CI; 0.70–1.63, $I^2$; 0.00%) (**S18 Fig**), having end-stage renal failure (pooled OR; 3.15, 95%CI; -0.07–6.38, $I^2$; 0.00%) (**S19 Fig**), being HIV positive (pooled OR; 2.26, 95%CI; 0.60–3.91, $I^2$; 43.47%) (**S20 Fig**), being high level INH resistance (pooled OR; 0.79, 5%CI; 0.36–1.21, $I^2$; 28.22%) (**S21 Fig**), taking INH in the initiation phase (pooled OR; 0.72, 95%CI; 0.33–1.11, $I^2$; 0.00%) (**S22 Fig**), taking STR in the initiation phase (pooled OR; 0.76, 95%CI; 0.15–1.37, $I^2$; 0.00%) (**S23 Fig**), taking FLQ in the initiation phase (pooled OR; 0.94, 95%CI; 0.48–1.39, $I^2$; 0.00%) (**S24 Fig**), taking PZA in the continuation phase (pooled OR; 0.87, 95%CI; 0.27–1.47, $I^2$; 0.00%) (**S25 Fig**), not culture converted after 2 months (pooled OR; 1.30, 95%CI; 0.59–2.00, $I^2$; 0.00%) (**S26 Fig**), and the presence cavity lesion in the chest radiograph (pooled OR; 1.23, 95%CI; 0.62–1.84, $I^2$; 0.00%) (**S27 Fig**) (**Table 2**).

## Discussion

Based on the pooled estimates, about one fifth of isoniazid mono-resistant TB patients had poor treatment outcomes and different factors are associated with this. The study findings of

**Table 2. The summary of the pooled on the poor treatment outcome and associated risk factors among patients with isoniazid mono-resistant tuberculosis per different categories.**

| Indicators | Number of studies | Pooled estimates | |
|---|---|---|---|
| | | Estimate (prevalence/OR), 95%CI | Heterogeneity $I^2$ |
| Successful treatment outcome | | | |
| Over all | 24 | 78% (74–83) | 94.02% |
| AFR | 4 | 67% (58–76) | 74.28% |
| AMR | 6 | 84% (78–90) | 87.66% |
| EMR | 2 | 75% (44–106) | 73.41% |
| EUR | 4 | 84% (77–91) | 91.21% |
| SEAR | 3 | 62% (56–69) | 13.74% |
| WPR | 5 | 82% (77–86) | 64.67% |
| Cure rate | 15 | 62% (50–73) | 96.91% |
| Complete rate | 14 | 34% (17–52) | 99.26% |
| Poor treatment outcome | | | |
| Over all | 23 | 22% (17–26) | 94.08% |
| AFR | 4 | 33% (24–42) | 74.28% |
| AMR | 6 | 16%(10–22) | 87.75% |
| EMR | 2 | 25% (-6-56) | 73.41% |
| EUR | 4 | 17% (11–22) | 85.06% |
| SEAR | 2 | 40% (34–45) | 0.00% |
| WPR | 5 | 18% (14–23) | 64.63% |
| Treatment failure | 16 | 5% (3–7) | 93.97% |
| Loss to follow-up | 18 | 12% (8–17) | 96.58% |
| Mortality | 23 | 6% (4–8) | 88.73% |
| Relapse after successful outcome | 8 | 1.7% (0.4–3.1) | n33.58% |
| Risk factors of poor treatment outcome | | | |
| Previous anti-TB treatment | 9 | **1.74 (1.15–2.33)** | 45.10% |
| Male sex | 9 | 1.34 (0.90–1.77) | 43.67% |
| Older age | 9 | 0.97 (0.62, 1.32) | 87.56% |
| Had HIV co-infection | 6 | 2.26 (0.60–3.91) | 43.47% |
| Smoking | 2 | 2.54 (0.89–4.20) | 13.69% |
| Had diabetes | 3 | 1.16 (0.70–1.63) | 0.00% |
| Had cancer | 2 | **3.53 (1.43–5.62)** | 0.00% |
| Had end stage renal disease | 2 | 3.15 (-0.07–6.38) | 0.00% |
| Being smear positive initially | 7 | **1.26 (1.08–1.43)** | 2.13% |
| Had high level INH resistance | 6 | 0.79 (0.38–1.21) | 28.22% |
| Took INH in the initiation phase | 2 | 0.72(0.33–1.11) | 0.00% |
| Took STR in the initiation phase | 2 | 0.76 (0.15–1.37) | 0.00% |
| Took FLQ in the initiation phase | 3 | 0.94 (0.48–1.39) | 0.00% |
| Took RIF in the continuation phase | 2 | **0.22 (0.04–0.41)** | 0.00% |
| Took PZA in the continuation phase | 2 | 0.87 (0.27–1.47) | 0.005 |
| Had extrapulmonary tuberculosis | 4 | **0.70 (0.55–0.85)** | 0.00% |
| Not culture converted after 2 months of treatment | 4 | 1.30 (0.59–2.00) | 0.00% |
| Took SLIDs | 2 | **0.54 (0.33–0.75)** | 0.00% |
| Had cavity lesion on the chest radiograph | 3 | 1.23 (0.62–1.84) | 0.00% |

AFR; African region, AMR; Region of the Americas, EMR; Eastern Mediterranean Region, EUR; European Region, SEAR; South Eastern region, WPR; West Pacific Region, HIV; Human Immunodeficiency Virus, INH; Isoniazid, RIF; Rifampicin, STR; Streptomycin; FLQ; Fluoroquinolones, PZA; Pyrazinamide; SLIDs; Second Line Injectable Drugs, OR; Odds Ratio

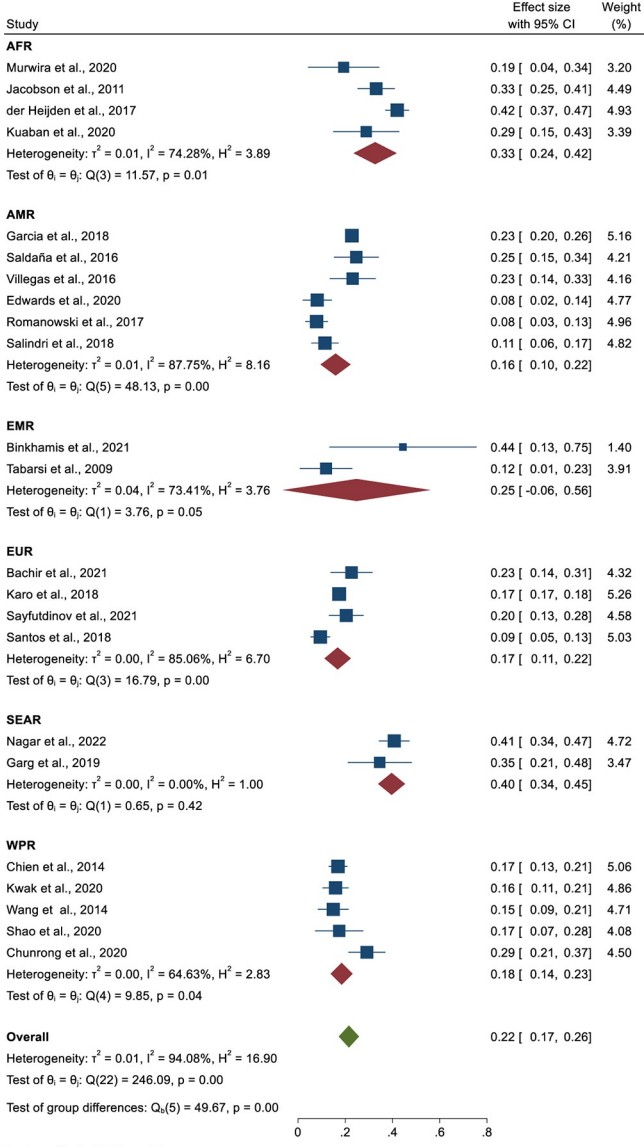

τ2; Tau (between-study variance), I²; I-squared heterogeneity statistic (variability between studies), H²; H-squared

heterogeneity statistic (variability between studies).

**Fig 4. Forest plot for the pooled poor treatment outcome rate among patients with isoniazid mono-resistant tuberculosis.**

this study revealed that the successful treatment rate among isoniazid mono-resistant TB patients was 79%. This finding is lower than the global average of the successful treatment outcome among drug-susceptible TB cases which was 85% and 86% for people newly enrolled on treatment in 2018 and in 2019, respectively [1, 2]. However, this is higher than MDR/RR-TB cases which was 59% based on the latest cohort [1], thus determining isoniazid resistant status for all bacteriologically confirmed TB cases may contribute for better treatment outcome and prevention of additional drug resistance. The successful treatment outcome among isoniazid mono-resistant TB cases had regional disparities, where better treatment success rate was

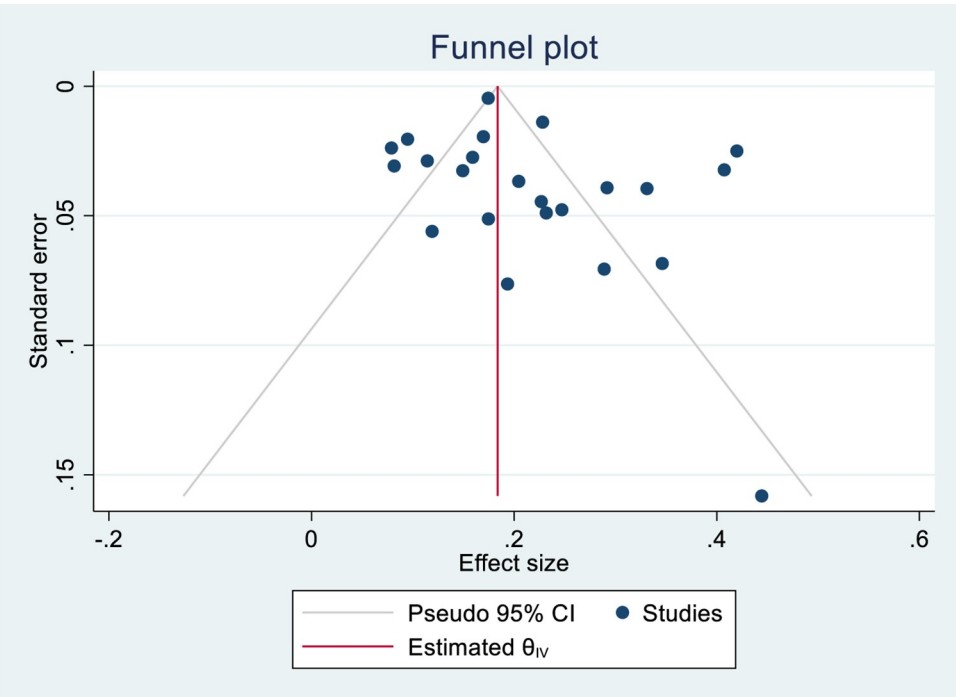

**Fig 5. Funnel plot for the pooled poor treatment outcome rate among patients with isoniazid mono-resistant tuberculosis.**

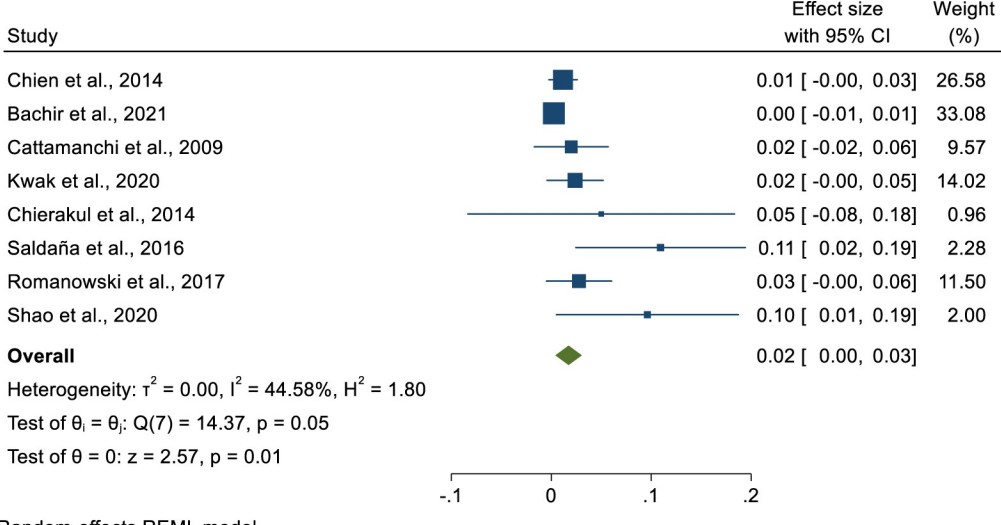

$\tau 2$; Tau (between-study variance), $I^2$; I-squared heterogeneity statistic (variability between studies), $H^2$; H-squared heterogeneity statistic (variability between studies).

**Fig 6. Forest plot for the pooled relapse rate among successfully treated patients with isoniazid mono-resistant tuberculosis.**

**Table 3. Meta-regression analysis of heterogeneity using sample size and publication year on poor treatment outcome.**

| Variable | Unadjusted model | | Adjusted model | |
|---|---|---|---|---|
| | Coefficient (95%CI) | P-value | Coefficient (95%CI) | P-value |
| Sample size | -5.76e-06 (-0.000035, 0.0000241) | 0.719 | -5.64e-06 (-0.0000357, 0.0000244) | 0.713 |
| Publication year | 0.0051572 (-0.0085616, 0.0188759) | 0.461 | .0052476 (-0.0087989, 0.0192942) | 0.464 |

noted from AMR, EUR, and WPR having a successful treatment outcome above 80%, while lower treatment outcome was noted in AFR and SEAR having 71% and 62%, respectively. This revealed the importance of taking regional and country specific interventions.

The pooled poor treatment outcome among isoniazid-mono resistant TB patients estimated in this study is higher compared to drug-susceptible TB patients at the global level [1, 2]. Thus, determining isoniazid resistance level for all bacteriologically confirmed TB cases is important. In developing countries there is a gap in addressing the universal access to DST. Besides, most of the countries are using GeneXpert for the simultaneous detection of TB and rifampicin resistance. This test determines only the drug resistance pattern to rifampicin. Thus, the isoniazid resistance level may be underestimated and may be treated as drug susceptible TB. This might have resulted with poor treatment outcomes and increasing drug resistance [2]. Based on the sub-group analysis, higher poor treatment outcome is noted in the SEAR. Likewise, based on the 2020 global TB report, lower MDR/RR-TB treatment success rate was noted in SEAR [2].

We have estimated the pooled proportion of relapse among successfully treated isoniazid mono-resistant TB cases. The finding revealed that two percent of those patients had a relapse that extends up to two years after treatment completion. This relapse rate is relatively lower than the 3.7% relapse rate in a pooled estimate among patients enrolled on DOTs program

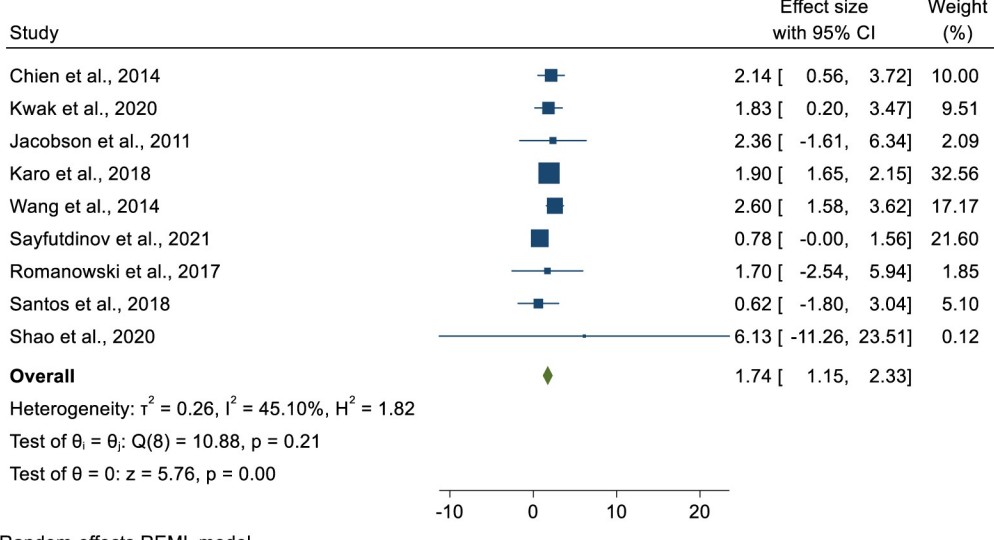

$\tau 2$; Tau (between-study variance), $I^2$; I-squared heterogeneity statistic (variability between studies), $H^2$; H-squared heterogeneity statistic (variability between studies).

**Fig 7. Forest plot for the association of previous TB treatment history with poor treatment outcome among isoniazid mono-resistant tuberculosis patients.**

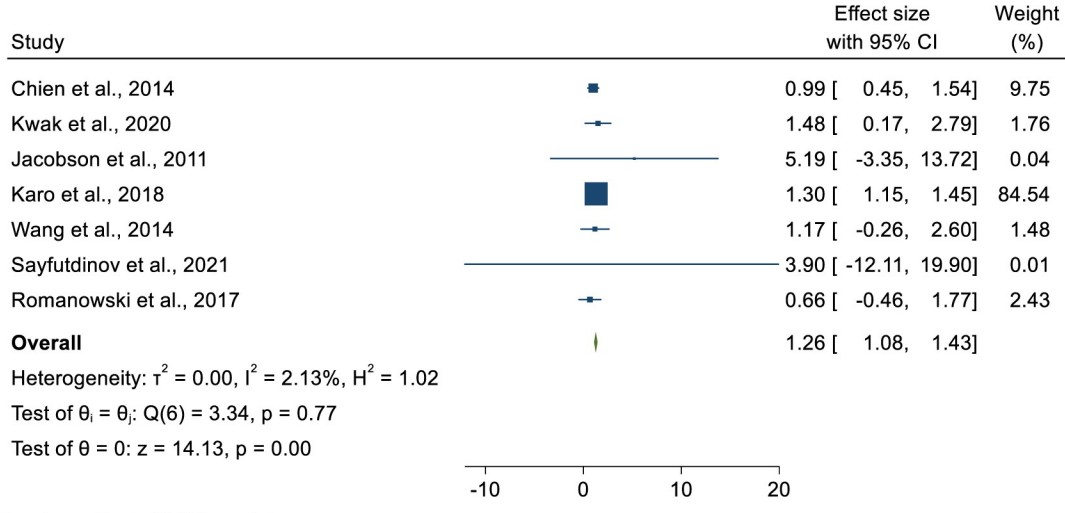

τ2; Tau (between-study variance), I$^2$; I-squared heterogeneity statistic (variability between studies), H$^2$; H-squared

heterogeneity statistic (variability between studies).

**Fig 8. Forest plot for the association of being initially smear positive with poor treatment outcome among isoniazid mono-resistant tuberculosis patients.**

[39]. The pooled estimate in our study might be affected because the time of follow-up was different among the studies.

In the current study, we have conducted a pooled estimate to assess the factors associated with poor treatment outcome in isoniazid mono-resistant TB cases. The study findings revealed that, those patients who had a previous TB treatment history had 1.74 times the odds to develop poor treatment outcome compared to new cases. Association of previous TB treatment history for developing unsuccessful treatment outcome in TB patients for both drug-susceptible and drug-resistant TB was reported in different studies [40–44]. This risk factor is not specific to isoniazid mono-resistant TB, rather it is associated with unfavorable TB treatment outcome in general. The other identified risk factor is being smear positive initially. Initially smear positive patients had 1.26 times the odds to develop poor treatment outcome compared to smear negatives. Smear positive TB patients had higher bacterial load in their sputum reflecting the severity of the disease. Likewise, a global pooled estimate revealed that drug-resistant TB patients who were smear positive at the baseline had 1.58 times the risk to die [41]. Besides, those isoniazid mono-resistant TB cases who had cancer comorbid had 3.53 times the odds to had poor treatment outcome compared to the counter parts. Similarly, in a previous study it was reported that the 12-months all-cause mortality during TB in patients with malignancy was as high as 20.56% [45]. Thus, those patients with comorbid conditions should be critically followed during treatment.

The findings of this study also revealed that those patients who took rifampicin in the continuation phase had lower risk to develop poor treatment outcome. Including rifampicin for treatment of isoniazid-mono resistant TB cases is important to shorten the treatment duration. Our study also revealed that taking SLIDs lowered the risk of poor treatment outcome. However, in patients with confirmed rifampicin-susceptible and isoniazid-resistant TB, it is not recommended to add injectable agents to the treatment regimen [46]. In addition, compared to

PTB cases EPTB cases had 45% reduced risk to develop poor treatment outcome which needs further studies. It is difficult to document treatment cure in EPTB cases. In two studies conducted in Ethiopia, EPTB was reported as the risk factor for unsuccessful treatment outcome [47, 48].

Finally, the findings of this study should be interpreted by considering the limitations. The study findings of this study was based on a limited number of studies (24 studies) with small sample size for the majority that might affected the pooled estimates. In addition, in the majority of the primary studies data were collected retrospectively that might have introduced selection bias. Besides, there is high heterogeneity and publication bias was detected for some parameters that might affect the true estimates. However, we have performed a stratified analysis and we also performed a trim and fill analysis for those pooled estimates that had a publication bias that validated the findings of this study.

## Conclusion

The findings of this study revealed that isoniazid mono-resistant TB patients had higher poor treatment outcome. The pooled estimates vary per geographical locations. Previous anti-TB treatment history, being smear positive initially, and having cancer were associated with poor treatment outcome in isoniazid mono-resistant TB patients. While, taking rifampicin in the continuation phase, taking SLIDs and having EPTB were associated with reduced risk of poor treatment outcome compared to their counter parts. Thus, determination of isoniazid resistance pattern for all bacteriological TB cases is critical to have successful treatment outcome.

## Supporting information

**S1 Table. Completed PRISMA 2009 checklist.**
(DOCX)

**S2 Table. Search engines.**
(DOCX)

**S3 Table. Quality assessment for the included studies in meta-analysis.**
(DOCX)

**S1 Fig. Forest plot for the complete rate among isoniazid mono-resistant tuberculosis patients.**
(DOCX)

**S2 Fig. Funnel plot for the complete rate among isoniazid mono-resistant tuberculosis patients.**
(DOCX)

**S3 Fig. Forest plot for the cure rate among isoniazid mono-resistant tuberculosis patients.**
(DOCX)

**S4 Fig. Funnel plot for the cure rate among isoniazid mono-resistant tuberculosis patients.**
(DOCX)

**S5 Fig. Forest plot for the treatment failure rate among isoniazid mono-resistant tuberculosis patients.**
(DOCX)

**S6 Fig. Funnel plot for the treatment failure rate among isoniazid mono-resistant tuberculosis patients.**
(DOCX)

**S7 Fig. Forest plot for the mortality rate among isoniazid mono-resistant tuberculosis patients.**
(DOCX)

**S8 Fig. Funnel plot for the mortality rate among isoniazid mono-resistant tuberculosis patients.**
(DOCX)

**S9 Fig. Forest plot for the lost to follow-up rate among isoniazid mono-resistant tuberculosis patients.**
(DOCX)

**S10 Fig. Funnel plot for the lost to follow-up rate among isoniazid mono-resistant tuberculosis patients.**
(DOCX)

**S11 Fig. Forest plot for the association of having cancer with poor treatment outcome among isoniazid mono-resistant tuberculosis patients.**
(DOCX)

**S12 Fig. Forest plot for the association of taking rifampicin in the continuation phase with poor treatment outcome among isoniazid mono-resistant tuberculosis patients.**
(DOCX)

**S13 Fig. Forest plot for the association of having extrapulmonary tuberculosis with poor treatment outcome among isoniazid mono-resistant tuberculosis patients.**
(DOCX)

**S14 Fig. Forest plot for the association of taking second-line injectable drugs with poor treatment outcome among isoniazid mono-resistant tuberculosis patients.**
(DOCX)

**S15 Fig. Forest plot for the association of being male with poor treatment outcome among isoniazid mono-resistant tuberculosis patients.**
(DOCX)

**S16 Fig. Forest plot for the association of older age with poor treatment outcome among isoniazid mono-resistant tuberculosis patients.**
(DOCX)

**S17 Fig. Forest plot for the association of smoking with poor treatment outcome among isoniazid mono-resistant tuberculosis patients.**
(DOCX)

**S18 Fig. Forest plot for the association of having diabetes with poor treatment outcome among isoniazid mono-resistant tuberculosis patients.**
(DOCX)

**S19 Fig. Forest plot for the association of having end stage renal disease with poor treatment outcome among isoniazid mono-resistant tuberculosis patients.**
(DOCX)

**S20 Fig. Forest plot for the association of being HIV positive with poor treatment outcome among isoniazid mono-resistant tuberculosis patients.**
(DOCX)

**S21 Fig. Forest plot for the association of having high-level isoniazid resistance with poor treatment outcome among isoniazid mono-resistant tuberculosis patients.**
(DOCX)

**S22 Fig. Forest plot for the association of taking isoniazid in the initiation phase with poor treatment outcome among isoniazid mono-resistant tuberculosis patients.**
(DOCX)

**S23 Fig. Forest plot for the association of taking streptomycin in the initiation phase with poor outcome among isoniazid mono-resistant tuberculosis patients.**
(DOCX)

**S24 Fig. Forest plot for the association of taking fluoroquinolones in the initiation phase with poor treatment outcome among isoniazid mono-resistant tuberculosis patients.**
(DOCX)

**S25 Fig. Forest plot for the association of taking pyrazinamide in the continuation phase with poor treatment outcome among isoniazid mono-resistant tuberculosis patients.**
(DOCX)

**S26 Fig. Forest plot for the association of not culture converted after 2 months' treatment of the initiation phase with poor treatment outcome among isoniazid mono-resistant tuberculosis patients.**
(DOCX)

**S27 Fig. Forest plot for the association of having cavity during chest radiograph with poor treatment outcome among isoniazid mono-resistant tuberculosis patients.**
(DOCX)

## Acknowledgments

We acknowledged the authors of the primary studies. We also acknowledge the Ethiopian Public Health Institute for access to internet searching.

## Author Contributions

**Conceptualization:** Ayinalem Alemu, Getachew Tollera.

**Data curation:** Ayinalem Alemu, Zebenay Workneh Bitew, Getu Diriba, Getachew Seid, Emebet Gashu.

**Formal analysis:** Ayinalem Alemu, Zebenay Workneh Bitew.

**Investigation:** Ayinalem Alemu, Getu Diriba, Getachew Seid, Emebet Gashu.

**Methodology:** Ayinalem Alemu, Zebenay Workneh Bitew, Kirubel Eshetu.

**Software:** Ayinalem Alemu, Zebenay Workneh Bitew.

**Writing – original draft:** Ayinalem Alemu.

**Writing – review & editing:** Ayinalem Alemu, Shewki Moga, Saro Abdella, Mesay Hailu Dangisso, Balako Gumi.

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
