## [Decision Letter · Decision Letter 0]

13 Feb 2023

PONE-D-22-32583Poor treatment outcome and associated risk factors among patients with isoniazid mono-resistant tuberculosis: a systematic review and meta-analysisPLOS ONE

Dear Dr. Alemu,

Thank you for submitting your manuscript to PLOS ONE. After careful consideration, we feel that it has merit but does not fully meet PLOS ONE’s publication criteria as it currently stands. Therefore, we invite you to submit a revised version of the manuscript that addresses the points raised during the review process.

We look forward to receiving your revised manuscript.

Kind regards,

Mohammad Mehdi Feizabadi, phd

Academic Editor

PLOS ONE

Journal Requirements:

Additional Editor Comments:

Despite the significance of the subject of this review, it needs major revision concerning the points raised by the reviewers.

Reviewers' comments:

Reviewer's Responses to Questions

**Comments to the Author**

1. Is the manuscript technically sound, and do the data support the conclusions?

Reviewer #1: Yes

Reviewer #2: Yes

2. Has the statistical analysis been performed appropriately and rigorously? 

Reviewer #1: No

Reviewer #2: No

3. Have the authors made all data underlying the findings in their manuscript fully available?

Reviewer #1: No

Reviewer #2: No

4. Is the manuscript presented in an intelligible fashion and written in standard English?

Reviewer #1: Yes

Reviewer #2: Yes

5. Review Comments to the Author

Reviewer #1: This study was done with the aim of "Poor treatment outcome an patients with isoniazid mono-resistant tuberculosis", which has important results, but in terms of writing, statistical analysis, the values of the tables need to be modified. Items requiring modification are available below

-METHOD : Inclusion and exclusion criteria, it should be added that there was no restriction on entering the study in terms of sample size.

-Method: To assess the presence of 174 publication bias, the funnel plot was inspected visually and Egger’s regression test was conducted 175 to ascertain the presence of publication bias.... This sentence is repeated twice in the method

-Line 176. Method: in the Egger’s regression test)….. Remove the parentheses

-Lin2 215: (P=0.1205). ..... p-values should be reported to three decimal places

-Line 270 : (INH, STR, FLQ, SLIDs) and continuation phase (RIF, PZA)….. Tb drugs that are mentioned for the first time in the text should be completed with the name

-line 302 : Based on the pooled estimates, about one fifth of isoniazid mono-resistant TB patients had poor treatment outcomes and different factors are associated with this……….

line 315: This study revealed that one among five (21%) isoniazid-mono resistant TB patients had poor…………..

These sentences are repeated twice

-TABLE 1: Cattamanchi et al., 2009 . The cells are empty

-TABLE 1: Study period....study period report should be same for some studies have reported by ‘month’ and ‘year’ while for other studies have written just ‘year’

-TABLE 1:Report of age groups should be same . For some studies have written "all" , some articles "all ( with median age )" and some other studies "median age ". it’s better that reported by min- max or mean ( sd)

-TABLE1: Chierakul: Celss are incomplete

-In table 1, the sum of the groups is not equal to the total, check again for example …

-TABLLE 1: 155 101 +50

-TABLE 1: der Heijden 407 235 +170

-Percentages check too please in table 1

-TABLE2

Treatment failure, Loss to follow-up ,Mortality= 16 18 23 … need check and correction

-TABLE 3: The writing format is not correct, while the content of the table is also ambiguous. Was the model based on linear regression? Or logistics? In this test, is the dependent variable of poor treatment with three subgroups ? (In this case, it is not possible to perform the logistic and linear regression test) or the type of outcome (poor and success). And why is the type of region or continent not included in the regression model?

-FIGURE 1: I2,H2,T2, …These indicators should be explained in the footnotes of all figures

-285 Line ant-TB treatment..... anti-TB treatment.

Reviewer #2: The subject of the research is very important and current. With the increasing need for new drugs for the treatment of tuberculosis, it is extremely important to determine the frequency of ripampicin resistance, the key of tuberculosis treatment.

The present manuscript describes a meta-analysis based on studies performed in the world.

Despite the importance of the topic addressed by the paper, many important points must be reviewed, specifically in the methodology:

1. Literature Searches and Search terms are incomplete. This is suboptimal for publication for systematic review. Search Embase is highly recommended. Please attach search terms that were used in each database as supplement for Data source and search strategies in the manuscript.

2. Quality assessment checklists are different for different types of studies, and researchers should choose the necessary checklist based on the type of article. Quality assessments for all included papers should be shown as a supplementary file.

3. The type and design of selected studies should be specified.

4. Another important issue is the considerable source of heterogeneity. High heterogeneity should be mentioned in the limitations section.

5. The method for selecting studies is not clear and needs further explanation.

6. The reasons for excluding the 7 articles (figure 1) should be stated.

7. Cohort studies cannot be combined with cross-sectional studies. All analyzes of these two types of studies should be performed separately and reported separately.

8. According to the global scale of the study, the number of included studies is very low and there are a large number of studies that should be included in the study by searching and re-screening.

6. PLOS authors have the option to publish the peer review history of their article (what does this mean?). If published, this will include your full peer review and any attached files.

Reviewer #1: No

Reviewer #2: No

---

## [Author Response · Author response to Decision Letter 0]

19 Feb 2023

Revisions based on the Editor’s and the reviewers’ comments and suggestions

Title: Poor treatment outcome and associated risk factors among patients with isoniazid mono-resistant tuberculosis: a systematic review and meta-analysis (PONE-D-22-32583)

Editor Comments and suggestions

We would like to thank the editor and the reviewers for giving pertinent comments and suggestions that improve the quality of the paper.

Journal Requirements:

• Response: Thank you, we have uploaded all required documents.

Additional Editor Comments:

Despite the significance of the subject of this review, it needs major revision concerning the points raised by the reviewers.

• Response: Thank you. We have revised the manuscript based on the important comments and suggestions given by both reviewers.

Reviewers' comments:

Reviewer's Responses to Questions

Comments to the Author

1. Is the manuscript technically sound, and do the data support the conclusions?

Reviewer #1: Yes

Reviewer #2: Yes

2. Has the statistical analysis been performed appropriately and rigorously?

Reviewer #1: No

Reviewer #2: No

• Response: Thank you for the critical review. Now we have revised the whole manuscript and updated the statistical analysis since some numbers are changed. The statistical analysis is now reviewed by a statistician. 

3. Have the authors made all data underlying the findings in their manuscript fully available?

Reviewer #1: No

Reviewer #2: No

• Response: Thank you for the pertinent review. All the supporting files are available within the manuscript and the supplementary files. 

4. Is the manuscript presented in an intelligible fashion and written in standard English?

Reviewer #1: Yes

Reviewer #2: Yes

5. Review Comments to the Author

Reviewer #1: This study was done with the aim of "Poor treatment outcome an patients with isoniazid mono-resistant tuberculosis", which has important results, but in terms of writing, statistical analysis, the values of the tables need to be modified. Items requiring modification are available below

-METHOD : Inclusion and exclusion criteria, it should be added that there was no restriction on entering the study in terms of sample size.

• Response: Thank you for the comment and suggestion. We included the sentence “There was no restriction on entering the study in terms of sample size.” in the revised manuscript. 

-Method: To assess the presence of 174 publication bias, the funnel plot was inspected visually and Egger’s regression test was conducted 175 to ascertain the presence of publication bias.... This sentence is repeated twice in the method

• Response: Thank you for the comment. Now, we revised it and avoided the unnecessary repetitions. 

-Line 176. Method: in the Egger’s regression test)….. Remove the parentheses

• Response: Thank you for the comment. Now, it is removed.

-Lin2 215: (P=0.1205). ..... p-values should be reported to three decimal places

• Response: Thank you for the comment. In the revised version, we presented the P-values in three decimal places.

-Line 270 : (INH, STR, FLQ, SLIDs) and continuation phase (RIF, PZA)….. Tb drugs that are mentioned for the first time in the text should be completed with the name

• Response: Thank you for the important comment. We presented their full name in their first presentation in the revised version. 

-line 302 : Based on the pooled estimates, about one fifth of isoniazid mono-resistant TB patients had poor treatment outcomes and different factors are associated with this……….

line 315: This study revealed that one among five (21%) isoniazid-mono resistant TB patients had poor…………..

These sentences are repeated twice

• Response: Thank you for the comment. We rephrased the repeated sentence. 

-TABLE 1: Cattamanchi et al., 2009 . The cells are empty

-TABLE1: Chierakul: Celss are incomplete

• Response: Thank you for the important comment. We put the sample size (137) for Cattamanchi et al., 2009 in the revised version, however the study only indicated the treatment completion rate the total successful treatment outcome including the cured cases and the poor treatment outcome (failure, death and lost to follow-up) are not indicated in the study. We mainly used this study to estimate the pooled prevalence of relapse in isoniazid mono-resistant tuberculosis patients. The Chierakul et al., 2014 didn’t specifically described the number of poor treatment outcome. In the current study, among the 25 studies, pooled treatment outcome is estimated using 23 studies excluding Cattamanchi et al., 2009 and Chierakul et al., 2014. Whereas, successful treatment outcome is estimated using 24 studies by excluding Cattamanchi et al., 2009. These are the reasons for the incomplete cells. 

-TABLE 1: Study period....study period report should be same for some studies have reported by ‘month’ and ‘year’ while for other studies have written just ‘year’

-TABLE 1:Report of age groups should be same . For some studies have written "all" , some articles "all ( with median age )" and some other studies "median age ". it’s better that reported by min- max or mean ( sd)

• Response: Thank you for this important comment. Unfortunately, the original studies did not describe the study period in similar way, some studies described the specific date, others indicated the month and some others described only the study year. We presented the study period based on the study period described in the primary studies. Likewise, the age of the study participants was described in different ways in the original studies. Thus, we presented the age groups already described by the original studies. We tried to present what is already described in the original studies. If there is any suggestion, we are happy. 

-In table 1, the sum of the groups is not equal to the total, check again for example …

-TABLLE 1: 155 101 +50

-TABLE 1: der Heijden 407 235 +170

-Percentages check too please in table 1

• Response: Thank you for these critical comments and suggestions. We have revised the figures of Table 1 and the gaps were arising from counting the non-evaluated patients in the denominator of three studies. Now, we revised the numbers and the percentages. We performed an analysis including pooled estimate forest plot, funnel plot, heterogeneity, publication bias and meta-regression analysis. All the numeral figures and forest plot/funnel plot figures are revised accordingly. 

-TABLE2

Treatment failure, Loss to follow-up ,Mortality= 16 18 23 … need check and correction

• Response: Thank you for the comment. The pooled treatment failure, lost to follow-up and mortality rate were estimated using 16, 18, and 23 studies, respectively. The figures described in Table 2 are correct. The forest plots are presented in the supplementary files (S5 fig for treatment failure, S9 fig for lost to follow-up and S7 fig for mortality).

-TABLE 3: The writing format is not correct, while the content of the table is also ambiguous. Was the model based on linear regression? Or logistics? In this test, is the dependent variable of poor treatment with three subgroups ? (In this case, it is not possible to perform the logistic and linear regression test) or the type of outcome (poor and success). And why is the type of region or continent not included in the regression model?

• Response: Thank you for the question and the comment. In Table 3, we have conducted a meta-regression analysis to assess the effect of sample size and publication year on the heterogeneity among studies that reported poor treatment outcome among isoniazid mono-resistant TB patients. It is a linear regression analysis where the poor treatment outcome estimate (proportion) is a dependent variable and the sample size and publication year were the independent variables. Based on the multivariable meta-regression model it is revealed that sample size (P=0.713) and publication year (P=0.464) did not significantly affected heterogeneity among studies. We described it in the result section under a title “Met-regression”. In Table 3, we presented the coefficient with its 95%CI in in the bi-variable and multivariable analysis. 

-FIGURE 1: I2,H2,T2, …These indicators should be explained in the footnotes of all figures

• Response: Thank you. We included the abbreviations in the footnotes of all figures. 

-285 Line ant-TB treatment..... anti-TB treatment.

• Response: Thank you for the question, it was a clerical problem. We changed previous anti-TB treatment to previous TB treatment history. 

Reviewer #2: The subject of the research is very important and current. With the increasing need for new drugs for the treatment of tuberculosis, it is extremely important to determine the frequency of ripampicin resistance, the key of tuberculosis treatment.

The present manuscript describes a meta-analysis based on studies performed in the world.

• Response: Thank you for the review. In this study, we conducted a study titled “Poor treatment outcome and associated risk factors among patients with isoniazid mono-resistant tuberculosis: a systematic review and meta-analysis”. The study mainly focused on isoniazid mono-resistant tuberculosis aimed at estimating the poor treatment outcome and its associated risk factors. 

Despite the importance of the topic addressed by the paper, many important points must be reviewed, specifically in the methodology:

1. Literature Searches and Search terms are incomplete. This is suboptimal for publication for systematic review. Search Embase is highly recommended. Please attach search terms that were used in each database as supplement for Data source and search strategies in the manuscript.

• Response: Thank you for the comment. Unfortunately, we don’t have access to Embase database. We have searched 5 data bases and 2 gray literature sources and we believe that relevant articles are accessed.

• The search strings for all the databases are included in the supplementary file (S2 Table). We have presented the search string for one database (PubMed) as an example in the method section of the main manuscript to limit the number of words in the manuscript. “The search string for PubMed was ("Treatment Outcome"[MeSH Terms] OR (("poverty"[MeSH Terms] OR "poverty"[All Fields] OR "poor"[All Fields]) AND ("Treatment Outcome"[MeSH Terms] OR ("treatment"[All Fields] AND "outcome"[All Fields]) OR "Treatment Outcome"[All Fields])) OR ("Treatment Outcome"[MeSH Terms] OR ("treatment"[All Fields] AND "outcome"[All Fields]) OR "Treatment Outcome"[All Fields])) AND (("isoniazid"[MeSH Terms] OR "isoniazid"[All Fields] OR "isoniazide"[All Fields]) AND "mono-resistant"[All Fields]) (S2 Table).”

2. Quality assessment checklists are different for different types of studies, and researchers should choose the necessary checklist based on the type of article. Quality assessments for all included papers should be shown as a supplementary file.

• Response: Thank you for the valuable comment. We have assessed the quality of each study using the Joanna Briggs Institute (JBI) critical appraisal tools for observational studies. As shown in the supplementary file (S3 Table), we have assessed the quality of the studies separately based on the study design used in each study. We have used the JBI checklist for cross-sectional, case control and cohort studies separately. 

3. The type and design of selected studies should be specified.

• Response: Thank you for the comment. The study design is specified for all studies as described in Table 1, column 5. 

4. Another important issue is the considerable source of heterogeneity. High heterogeneity should be mentioned in the limitations section.

• Response: Thank you for the comment. We have mentioned the potential impact of high heterogeneity as one limitation of the study. It is described in the limitation section. “Besides, there is high heterogeneity and publication bias was detected for some parameters that might affect the true estimates.”

5. The method for selecting studies is not clear and needs further explanation.

• Response: Thank for the comment. We put this paragraph in the study selection procedure section. “We have followed a step-wise approach to select the eligible studies. Primarily, all the studies identified from the whole search were exported to EndNote X8 citation manager, and we have removed the duplicates. In the next step, we have screened the articles by title and abstract. Then, full-text assessment was conducted for the remaining articles. Finally, we have included the articles that passed the full-text review in the final analysis. The article selection procedure was conducted by two independent authors (GD, GS) using pre-defined criteria that considered study subjects, study designs, quality, and outcome (Fig 1).” 

• We selected the studies using the PICOS criteria. Finally, we put an inclusion and exclusion criteria to select studies to be included in the final analysis. 

6. The reasons for excluding the 7 articles (figure 1) should be stated.

• Response: Thank you for the comment. We excluded 4 studies (Fig 1). Three studies were reviews (are not primary studies) and one study was excluded because it did not describe the specific outcome. 

7. Cohort studies cannot be combined with cross-sectional studies. All analyzes of these two types of studies should be performed separately and reported separately.

• Response: Thank you for the comment and suggestion. In the current study, we estimated the pooled prevalence of poor treatment outcome among isoniazid mono-resistant TB patients. Since both cross-sectional and cohort studies are observational studies, we merged together to estimate the pooled prevalence. We performed a sub-group analysis and there is no significant difference among the study designs. 

8. According to the global scale of the study, the number of included studies is very low and there are a large number of studies that should be included in the study by searching and re-screening.

• Response: Thank you for the important comment. There is limited evidence that described the treatment outcome of isoniazid mono-resistant tuberculosis and associated risk factors. Since our focus is on isoniazid mono-resistant tuberculosis, we identified limited number of studies. We described the small number of studies as a limitation of the study. “The study findings of this study was based on a limited number of studies (24 studies) with small sample size for the majority that might affected the pooled estimates.”

Kind regards, 

Ayinalem Alemu 

Ethiopian Public Health Institute 

National Tuberculosis Reference Laboratory, 

P.O. Box 1242, Addis Ababa, Ethiopia. 

Email: ayinalemal@gmail.com

 Tell: +251912366676

---

## [Decision Letter · Decision Letter 1]

18 Apr 2023

PONE-D-22-32583R1Poor treatment outcome and associated risk factors among patients with isoniazid mono-resistant tuberculosis: a systematic review and meta-analysisPLOS ONE

Dear Dr. Alemu,

Thank you for submitting your manuscript to PLOS ONE. After careful consideration, we feel that it has merit but does not fully meet PLOS ONE’s publication criteria as it currently stands. Therefore, we invite you to submit a revised version of the manuscript that addresses the points raised during the review process.

Please submit your revised manuscript by Jun 02 2023 11:59PM. If you will need significantly more time to complete your revisions, please reply to this message or contact the journal office at plosone@plos.org. Please include the following items when submitting your revised manuscript:A rebuttal letter that responds to each point raised by the academic editor and reviewer(s). You should upload this letter as a separate file labeled 'Response to Reviewers'.A marked-up copy of your manuscript that highlights changes made to the original version. You should upload this as a separate file labeled 'Revised Manuscript with Track Changes'.An unmarked version of your revised paper without tracked changes. You should upload this as a separate file labeled 'Manuscript'.If applicable, we recommend that you deposit your laboratory protocols in protocols.io to enhance the reproducibility of your results. Protocols.io assigns your protocol its own identifier (DOI) so that it can be cited independently in the future. For instructions see: https://journals.plos.org/plosone/s/submission-guidelines#loc-laboratory-protocols. Additionally, PLOS ONE offers an option for publishing peer-reviewed Lab Protocol articles, which describe protocols hosted on protocols.io. Read more information on sharing protocols at https://plos.org/protocols?utm_medium=editorial-email&utm_source=authorletters&utm_campaign=protocols.

We look forward to receiving your revised manuscript.

Kind regards,

Frederick Quinn

Academic Editor

PLOS ONE

Journal Requirements:

Reviewers' comments:

Reviewer's Responses to Questions

**Comments to the Author**

1. If the authors have adequately addressed your comments raised in a previous round of review and you feel that this manuscript is now acceptable for publication, you may indicate that here to bypass the “Comments to the Author” section, enter your conflict of interest statement in the “Confidential to Editor” section, and submit your "Accept" recommendation.

Reviewer #1: All comments have been addressed

Reviewer #2: (No Response)

2. Is the manuscript technically sound, and do the data support the conclusions?

Reviewer #1: Yes

Reviewer #2: (No Response)

3. Has the statistical analysis been performed appropriately and rigorously? 

Reviewer #1: Yes

Reviewer #2: (No Response)

4. Have the authors made all data underlying the findings in their manuscript fully available?

Reviewer #1: Yes

Reviewer #2: (No Response)

5. Is the manuscript presented in an intelligible fashion and written in standard English?

Reviewer #1: Yes

Reviewer #2: (No Response)

6. Review Comments to the Author

Reviewer #1: 1. Line 270: please write ….. isoniazid (INH), streptomycin(STR), fluoroquinolones(FLQ), second-line injectable drugs (SLIDs) and continuation phase(rifampicin(RIF), pyrazinamide (PZA)).

2. Table 1 : Its need you explain in footnote of table1

Cattamanchi et al., 2009*

* the study only indicated the treatment completion rate the total successful treatment outcome including the cured cases and the poor treatment outcome (failure, death and lost to follow-up) are not indicated in the study.

3.Table 1 : Chien et al., 2014 : All (Median age was 64 years) . What does "all" mean? All age groups? 64 years seems to be high for the middle age, please recheck. For all studies that include all age groups, please add the word "all age groups". For example Binkhamis et al., 202…. all age groups ( range: 1-90 years).

4.(Median zge was 41 years)……. Median age was 41 years

Reviewer #2: (No Response)

7. PLOS authors have the option to publish the peer review history of their article (what does this mean?). If published, this will include your full peer review and any attached files.

Reviewer #1: No

Reviewer #2: No

---

## [Author Response · Author response to Decision Letter 1]

19 Apr 2023

Revisions based on the Editor’s and the reviewers’ comments and suggestions

Title: Poor treatment outcome and associated risk factors among patients with isoniazid mono-resistant tuberculosis: a systematic review and meta-analysis (PONE-D-22-32583R1)

Journal Requirements:

• Response: Thank you, the reference list is complete and correct. A retracted paper is not cited. 

Reviewers' comments:

Reviewer's Responses to Questions

Comments to the Author

1. If the authors have adequately addressed your comments raised in a previous round of review and you feel that this manuscript is now acceptable for publication, you may indicate that here to bypass the “Comments to the Author” section, enter your conflict of interest statement in the “Confidential to Editor” section, and submit your "Accept" recommendation.

Reviewer #1: All comments have been addressed

Reviewer #2: (No Response)

• Response: Thank you for the pertinent review.

2. Is the manuscript technically sound, and do the data support the conclusions?

Reviewer #1: Yes

Reviewer #2: (No Response)

• Response: Thank you for the pertinent review.

3. Has the statistical analysis been performed appropriately and rigorously?

Reviewer #1: Yes

Reviewer #2: (No Response)

• Response: Thank you for the pertinent review.

4. Have the authors made all data underlying the findings in their manuscript fully available?

Reviewer #1: Yes

Reviewer #2: (No Response)

• Response: Thank you for the pertinent review.

5. Is the manuscript presented in an intelligible fashion and written in standard English?

Reviewer #1: Yes

Reviewer #2: (No Response)

• Response: Thank you for the pertinent review.

6. Review Comments to the Author

Reviewer #1: 

1. Line 270: please write ….. isoniazid (INH), streptomycin(STR), fluoroquinolones(FLQ), second-line injectable drugs (SLIDs) and continuation phase(rifampicin(RIF), pyrazinamide (PZA)).

• Response: Thank you for the comment. It is revised accordingly.

2. Table 1 : Its need you explain in footnote of table1

Cattamanchi et al., 2009*

* the study only indicated the treatment completion rate the total successful treatment outcome including the cured cases and the poor treatment outcome (failure, death and lost to follow-up) are not indicated in the study.

• Response: Thank you for the valuable comment and suggestion. We revised it based on the given suggestion.

3.Table 1 : Chien et al., 2014 : All (Median age was 64 years) . What does "all" mean? All age groups? 64 years seems to be high for the middle age, please recheck. For all studies that include all age groups, please add the word "all age groups". For example Binkhamis et al., 202…. all age groups ( range: 1-90 years).

• Response: Thank you for the valuable comments. We replaced the word “All” with the phrase “all age groups”. The median age described in the Chien et al., 2014 study is 64.0 years as indicated in page number 61 of their study. 

4.(Median zge was 41 years)……. Median age was 41 years

• Response: Thank you, we revised it.

Reviewer #2: (No Response

Kind regards, 

Ayinalem Alemu 

Ethiopian Public Health Institute 

National Tuberculosis Reference Laboratory

P.O. Box 1242, Addis Ababa, Ethiopia. 

Email: ayinalemal@gmail.com

Tell: +251912366676

---

## [Decision Letter · Decision Letter 2]

11 May 2023

Poor treatment outcome and associated risk factors among patients with isoniazid mono-resistant tuberculosis: a systematic review and meta-analysis

PONE-D-22-32583R2

Dear Dr. Alemu,

We’re pleased to inform you that your manuscript has been judged scientifically suitable for publication and will be formally accepted for publication once it meets all outstanding technical requirements.

Kind regards,

Frederick Quinn

Academic Editor

PLOS ONE

Additional Editor Comments (optional):

Reviewers' comments:

Reviewer's Responses to Questions

**Comments to the Author**

1. If the authors have adequately addressed your comments raised in a previous round of review and you feel that this manuscript is now acceptable for publication, you may indicate that here to bypass the “Comments to the Author” section, enter your conflict of interest statement in the “Confidential to Editor” section, and submit your "Accept" recommendation.

Reviewer #1: All comments have been addressed

2. Is the manuscript technically sound, and do the data support the conclusions?

Reviewer #1: Yes

3. Has the statistical analysis been performed appropriately and rigorously? 

Reviewer #1: Yes

4. Have the authors made all data underlying the findings in their manuscript fully available?

Reviewer #1: Yes

5. Is the manuscript presented in an intelligible fashion and written in standard English?

Reviewer #1: Yes

6. Review Comments to the Author

Reviewer #1: This study with title of Poor treatment outcome and associated risk factors among patients with isoniazid mono-resistant tuberculosis: a systematic review and meta-analysis

The manuscript has been edited, It is well written and has interesting and valuable content, I think it is worth publishing

7. PLOS authors have the option to publish the peer review history of their article (what does this mean?). If published, this will include your full peer review and any attached files.

Reviewer #1: **Yes: **Please do not sign this review on behalf of another person

---

## [Editor Report · Acceptance letter]

11 Jul 2023

PONE-D-22-32583R2 

Poor treatment outcome and associated risk factors among patients with isoniazid mono-resistant tuberculosis: a systematic review and meta-analysis 

Dear Dr. Alemu:

I'm pleased to inform you that your manuscript has been deemed suitable for publication in PLOS ONE. Congratulations! Your manuscript is now with our production department. 

Kind regards, 

on behalf of

Dr. Frederick Quinn 

Academic Editor

PLOS ONE